# Expression of GP88 (Progranulin) Protein Is an Independent Prognostic Factor in Prostate Cancer Patients

**DOI:** 10.3390/cancers11122029

**Published:** 2019-12-16

**Authors:** Amer Abdulrahman, Markus Eckstein, Rudolf Jung, Juan Guzman, Katrin Weigelt, Ginette Serrero, Binbin Yue, Carol Geppert, Robert Stöhr, Arndt Hartmann, Bernd Wullich, Sven Wach, Helge Taubert, Verena Lieb

**Affiliations:** 1Department of Urology and Pediatric Urology, Universitätsklinikum Erlangen, Friedrich-Alexander-Universität Erlangen-Nürnberg, 91054 Erlangen, Germany; Amer.Abdulrahman@uk-erlangen.de (A.A.); Juan.Guzman@uk-erlangen.de (J.G.); Katrin.Weigelt@uk-erlangen.de (K.W.); Bernd.Wullich@uk-erlangen.de (B.W.); sven.wach@uk-erlangen.de (S.W.); Verena.Lieb@uk-erlangen.de (V.L.); 2Department of Pathology, Universitätsklinikum Erlangen, Friedrich-Alexander-Universität Erlangen-Nürnberg, 91054 Erlangen, Germany; Markus.Eckstein@uk-erlangen.de (M.E.); Rudolf.Jung@uk-erlangen.de (R.J.); carol.geppert@uk-erlangen.de (C.G.); robert.stoehr@uk-erlangen.de (R.S.); arndt.hartmann@uk-erlangen.de (A.H.); 3A&G Pharmaceutical Inc., Columbia, MD 21045, USA; gserrero@agpharma.com; 4Program in Oncology, University of Maryland Greenebaum Comprehensive Cancer Center, Baltimore, MD 21201, USA; byue@agpharma.com

**Keywords:** GP88, progranulin, prostate cancer, gleason score, prognosis

## Abstract

Prostate cancer, the second most common cancer, is still a major cause of morbidity and mortality among men worldwide. The expression of the survival and proliferation factor progranulin (GP88) has not yet been comprehensively studied in PCa tumors. The aim of this study was to characterize GP88 protein expression in PCa by immunohistochemistry and to correlate the findings to the clinico-pathological data and prognosis. Immunohistochemical staining for GP88 was performed by TMA with samples from 442 PCa patients using an immunoreactive score (IRS). Altogether, 233 cases (52.7%) with negative GP88 staining (IRS < 2) and 209 cases (47.3%) with positive GP88 staining (IRS ≥ 2) were analyzed. A significant positive correlation was found for the GP88 IRS with the PSA value at prostatectomy and the cytoplasmic cytokeratin 20 IRS, whereas it was negatively associated with follow-up times. The association of GP88 staining with prognosis was further studied by survival analyses (Kaplan–Meier, univariate and multivariate Cox’s regression analysis). Increased GP88 protein expression appeared as an independent prognostic factor for overall, disease-specific and relapse-free survival in all PCa patients. Interestingly, in the subgroup of younger PCa patients (≤65 years), GP88 positivity was associated with a 3.8-fold (*p* = 0.004), a 6.0-fold (*p* = 0.008) and a 3.7-fold (*p* = 0.003) increased risk for death, disease-specific death and occurrence of a relapse, respectively. In the PCa subgroup with negative CK20 staining, GP88 positivity was associated with a 1.8-fold (*p* = 0.018) and a 2.8-fold increased risk for death and disease-specific death (*p* = 0.028). Altogether, GP88 protein positivity appears to be an independent prognostic factor for PCa patients.

## 1. Introduction

Prostate cancer (PCa) is the second most common cancer, occupying position five of the tumor-associated deaths in men worldwide [1]. As such, prostate cancer is a significant public health burden and a major cause of morbidity and mortality among men worldwide. An additional problem is a continuing increase in especially early onset PCa in Europe and America [2]. PCa represents a group of histologically and molecularly heterogeneous diseases with variable clinical courses. In accordance with the recent knowledge of their clinico-pathologies and genetics, the World Health Organization (WHO) classification of prostatic cancers has been revised [3]. Five grade groups were established, i.e., grade group 1: Gleason score (GS) ≤ 6; grade group 2: GS 3 + 4 = 7; grade group 3: GS 4 + 3 = 7; grade group 4: GS 8; and grade group 5: GS 9–10. In addition, molecular classification of several tumors, including prostate cancer, at the RNA level revealed a luminal-like and basal-like gene expression pattern. For prostate cancer, this classification is associated with prognosis and response to androgen deprivation therapy [4,5]. In this study, we applied CK20 as a proxy marker for a luminal-like gene expression pattern. However, such a classification for PCa based on protein expression does not exist yet. There are several biomarkers and biomarker assays on the market [6,7,8,9], but none of them have really arrived in the urologic clinic yet. Therefore, there is still a need to search for compelling biomarkers in PCa.

Progranulin (also known as GP88, PCDGF, Acrogranin, Proepithelin) is an 88-kDa glycoprotein and the largest member of the granulin/epithelin family, first isolated from human bone marrow [10]. It acts as an autocrine proliferation and survival factor for several cancer types [11]. Increased expression of the 88-kDa glycoprotein (GP88/PGRN) has been reported, e.g., in breast cancer, brain tumors, ovarian cancer, renal carcinoma, bladder cancer, non-small cell lung cancer, and hematological cancers [12,13,14,15,16]. Concerning prostate cancer, Pan et al. reported that GP88 expression occurred during the development of prostatic intraepithelial neoplasia (PIN). It was not or only weakly expressed in normal prostate tissues, but both the intensity and the fraction of cells expressing GP88 were significantly greater in invasive cancer [17]. In vitro GP88 promotes cell growth, migration, invasion and anchorage-independent growth in prostate cancer cells [18]. Recently, we investigated the circulating levels of GP88 in the serum of prostate cancer patients and showed that all PCa patients and particularly younger PCa patients with a low serum GP88 level had a significantly better overall survival compared with that of patients with higher serum GP88 levels [19]. In the present study, we examined GP88 protein expression in PCa tissue and its correlation with clinico-pathological data and the prognosis of PCa patients.

## 2. Results

### 2.1. GP88 Expression and Correlation with Clinico-Pathological Parameters and the Expression of Selected Proteins

A cohort of 442 PCa patients was evaluated for their GP88 protein and cytokeratin 20 (CK20) expression by immunohistochemistry (IHC).

GP88 protein expression was detected in the cytoplasm and assessed with an IRS score, as described in the Methods section. CK20 expression was also assessed with an IRS score. Figure 1 provides representative photomicrographs of GP88- and CK20-stained biospecimens. We detected 233 cases (52.7%) with negative GP88 staining (IRS < 2) and 209 cases (47.3%) with positive GP88 staining (IRS ≥ 2). There were 314 cases (71%) that were CK20 negative (IRS < 2) and 128 cases (29%) that were CK20 positive (IRS ≥ 2) (Table 1 and Table 2; Appendix A).

Next, we tested whether GP88 staining was associated with clinico-pathological parameters by correlation tests (Spearman’s bivariate correlation test). The clinico-pathological data of the PCa patients are summarized in Table 1.

There was no correlation of the GP88 IRS with age, Gleason score or tumor size (Appendix A). A significant positive correlation was found for the GP88 IRS with the PSA value at prostatectomy (r_s_ = 0.201; *p* < 0.001) and cytoplasmic cytokeratin 20 (CK20) IRS (r_s_ = 0.31; *p* < 0.001; Appendix A). A negative association with follow-up time periods for OS and DSS (r_s_ = −1.51; *p* = 0.001) and for RFS (r_s_ = −1.49; *p* = 0.002) was detected (Appendix A).

When grouping GP88 IRS (IRS < 2 vs. IRS ≥ 2) and the PSA value at prostatectomy (<4 ng/mL vs. ≥4 ng/mL) or cytoplasmic CK20 IRS (IRS < 2 vs. IRS ≥ 2) in a cross table analysis, the distributions for the PSA value at prostatectomy (*p* = 0.001) and for cytoplasmic CK20 IRS (*p* < 0.001) were significantly different in both GP88 groups (Appendix A). The association of the GP88 IRS groups with prognosis was further tested by the Kaplan–Meier analysis as well as by univariate and multivariate Cox’s regression analyses.

### 2.2. Association of GP88 Protein Expression and Survival

There was an association of GP88 staining with OS (*p* = 0.002), DSS (*p* = 0.018), and RFS (*p* = 0.040) observed by the Kaplan–Meier analysis (Table 3; Figure 2). The mean overall, disease-specific and relapse-free survival times were 164.9 months, 204.3 months and 186.4 months, respectively, for GP88-positive patients and 195.8 months, 231.1 months and 218.1 months for GP88-negative patients. The univariate Cox’s regression analysis (Table 4) revealed that GP88 positivity was associated with a 1.9-fold, 2.5-fold and 1.7-fold increased risk of death (*p* = 0.003), tumor-related death (*p* = 0.021) and relapse occurrence (*p* = 0.043), respectively. Multivariate Cox’s regression analysis adjusted for the Gleason score, tumor stage and age (Table 4) revealed that GP88 staining was an independent prognostic factor of OS (HR = 1.8; *p* = 0.011), DSS (HR = 2.4; *p* = 0.039) and RFS (HR = 1.7; *p* = 0.046), respectively. Next, we performed a subgroup analysis for tumor stage, Gleason score, patient age and CK20 staining.

### 2.3. Association of GP88 Protein Expression and Survival Stratified by Tumor Stage

We were interested in examining whether there were differences in prognosis associated with GP88 staining within the two tumor stage groups (pT2 vs. pT3+4). There was a difference in OS and DSS in the pT2 group (Table 3 and Table 4) but not in the pT3+4 group (data not shown). Moreover, no difference in both groups for RFS was detected (data not shown). In the pT2 group, a mean survival of 159.9 months and a mean disease-specific survival of 196.1 months were found for the GP88-positive patients in comparison to a mean survival of 193.4 months and 220.7 months for GP88-negative patients (Table 3; Appendix A). The univariate Cox’s regression analysis showed that GP88 positivity was associated with a 2.1-fold increased risk of death (*p* = 0.022, Table 4) and a 4.6-fold increased risk of tumor-related death (*p* = 0.030; Table 4) in the pT2 group. The multivariate Cox’s regression analysis (adjusted for the Gleason score and age) indicated that GP88 positivity was an independent prognostic factor for OS in the pT2 group (HR = 1.9; *p* = 0.043; Table 4).

### 2.4. Association of GP88 Protein Expression and Survival Stratified by Gleason Score

The Gleason score (GS) separates the PCa patients into five groups: GS6, GS7a (GS3+4), GS7b (GS4+3), GS8 and GS9-10. Differences in prognosis between GP88-positive and GP88-negative patients were detected for OS and DSS only in the GS7b group and for RFS in the GS8 group. All GP88-negative patients (N = 11) in the GS7b group (N = 32) survived (mean OS: *p* = 0.004 and mean DSS: *p* = 0.029; Table 3; Appendix A). In the GS8 group, patients with GP88 positivity survived relapse-free for 99.0 months, whereas those with GP88 negativity survived 201.8 months (*p* = 0.008; Table 3; Appendix A). Univariate Cox’s regression analysis revealed that GP88-positive patients had a 10.2-fold increased risk for relapse compared with GP88-negative patients (*p* = 0.033; Table 4). In the multivariate Cox’s regression analysis (adjusted for tumor stage and age), GP88 staining was not an independent prognostic factor (Table 4).

### 2.5. Association of GP88 Protein Expression and Survival Stratified by Patient’s Age

Patients were separated into two groups by their median age of 65 years (≤65 years vs. >65 years). Differences in prognosis between GP88-positive and GP88-negative patients were detected by the Kaplan–Meier analysis only in the younger group for OS (*p* = 0.001), DSS (*p* = 0.003) and RFS (*p* = 0.004). GP88-positive patients survived on average 169.5 months, disease-specific 186.1 months and relapse-free 153.1 months, whereas the GP88-negative patients’ survival was 227.3 months overall, 237.1 months disease-specific and 222.8 months relapse-free survival (Table 3; Appendix A). The univariate analysis showed that GP88-positive patients who were ≤ 65 years old had a 3.6-fold increased risk of death (*p* = 0.002), a 5.5-fold increased risk of disease-specific death (*p* = 0.006) and a 2.9-fold increased risk of relapse (*p* = 0.005; Table 4) compared to those of GP88-negative patients in the same age group. In the multivariate analysis adjusted for tumor stage and Gleason score, GP88 staining appeared to be an independent prognostic factor for OS (*p* = 0.004), DSS (*p* = 0.008) and RFS (*p* = 0.003) in patients 65 years old and younger. GP88-positive patients had a 3.8-fold increased risk for death, a 6-fold increased risk for disease-specific death and a 3.7-fold increased risk for relapse (Table 4).

### 2.6. Association of GP88 Protein Expression and Survival Stratified by CK20 Staining

We applied IHC for cytokeratin 20 (CK20) and separated the patients into two groups with negative or positive CK20 staining (IRS < 2 vs. IRS ≥ 2; Table 1 and Table 2). However, both groups did not differ in OS, DSS or RFS (data not shown).

Differences in survival between the GP88-positive and GP88-negative groups after stratification based on CK20 staining (negative: IRS < 2 and positive: IRS ≥ 2) were only detected in the CK20-negative group for OS (*p* = 0.009) and DSS (*p* = 0.005) but not for RFS (Table 3; Appendix A).

GP88-positive patients survived on average 162.9 months, disease-specific 197.5 months, whereas GP88-negative patients survived on average 186.8 months and disease-specific 216.1 months. In the univariate Cox’s regression analysis, the GP88-positive patients had a 1.9-fold increased risk of overall death (*p* = 0.010) and a 3.2-fold increased risk of disease-specific death (*p* = 0.007; Table 4). In the multivariate Cox’s regression analysis, the GP88-positive patients showed a 1.8-fold increased risk of overall death (*p* = 0.018) and a 2.8-fold increased risk of disease-specific death (*p* = 0.028; Table 4). Altogether, GP88 positivity remained an independent prognostic factor for OS and DSS.

Next, we combined GP88 and CK20 staining in four groups. Group 0 corresponds to cases with both markers negative; group 1: CK20 positive and GP88 negative; group 2: CK20 negative and GP88 positive; and group 3: both markers are positive. As expected, the groups with GP88 positivity (groups 2 and 3) had a poorer prognosis, but surprisingly, CK20 expression also affected prognosis. The groups with CK20 positivity had a better prognosis than the CK20-negative groups within the same GP88 groups (Figure 3). The mean overall survival and disease-specific survival were 186.8 months and 216.1 months in group 0, 217.4 months and 237.5 months in group 1, 162.3 months and 197.5 months in group 2, and 168.5 months and 208.4 months in group 3 (*p* = 0.005 and *p* = 0.015; Figure 3; Appendix A). In the univariate and multivariate Cox’s regression analyses, there was only an association between the GP88/CK20 staining groups and OS but not DSS (Appendix A). Group 2 (GP88+/CK20-) showed the worst survival, both by univariate (HR = 4.9; *p* = 0.008) and by multivariate analysis (HR = 6.0; *p* = 0.014), and group 3 (GP88+/CK20+) presented the second worst survival by univariate (HR = 3.7; *p* = 0.035) and multivariate analysis (HR = 4.5; *p* = 0.045) when compared with the survival of group 1 (GP88-/CK20+).

## 3. Discussion

In this study, GP88 protein expression was analyzed in tumors from 442 PCa patients and was correlated with clinico-pathological parameters to examine its prognostic value. The expression of GP88 protein was positively correlated with the PSA value at prostatectomy and cytoplasmic cytokeratin 20 (CK20) staining; moreover, increased GP88 in tumor tissue was a prognostic factor because it was correlated with shorter follow-up times for overall, disease-specific and relapse-free survival. Our previous study investigating the serum GP88 levels in prostate cancer patients [19] reported that serum GP88 levels were different between tumor stage and age groups; i.e., low GP88 levels were more often observed in lower tumor stage groups (pT1/pT2) and in the early age group (≤66 years), and vice versa, increased GP88 levels were more often observed in the higher tumor stage group (pT3/pT4) and in the older age group (>66 years). In the current study, GP88 expression was different in PSA level groups, i.e., low GP88 levels were more often found in the low PSA group (<4 ng/mL), and vice versa, increased GP88 levels were more often found in the higher PSA group (≥4 ng/mL). These results show that GP88 levels/GP88 protein expression in serum and in tumor tissue are correlated with different clinico-pathological parameters, suggesting possibly different roles in tumor biology.

We showed, for the first time, that increased GP88 protein expression (IRS ≥ 2) in PCa tumor tissues is an independent prognostic marker for shorter OS, DSS and RFS in PCa patients. Previously, we reported that serum GP88 expression was an independent prognostic factor for OS in PCa patients [19]. Interestingly, we show here that GP88 protein expression appeared to be an independent prognostic factor for OS, DSS and RFS only in the younger patient group (≤65 years) but not in the older patient group (>65 years). Again, this supports our previous finding that in younger PCa patients, high serum GP88 levels were associated with shorter OS of PCa patients by the univariate Cox’s regression analysis. The finding for younger PCa patients is of special interest since there is a continuing increase in early-onset PCa in Europe and America [2]. In addition, we found a positive correlation between GP88 and CK20 staining. Interestingly, only in the CK20-negative group (IRS < 2) was GP88 positivity associated independently with shorter OS and DSS. When the CK20 and GP88 staining results were combined, the group with CK20-negative and GP88-positive staining had the worst prognosis in OS and DSS, whereas patients with CK20 positivity and GP88 negativity had the best OS and DSS. Cytokeratin 20 at the RNA level is considered a marker for luminal-like subtypes of cancers, including prostate cancer. Cancers with cytokeratin positivity generally show a better prognosis than cases with low cytokeratin mRNA expression that belong to the basal-like subtype. At the protein level, cytokeratin 20 is not routinely used as a marker for luminal-like subtypes in prostate cancer. We did not observe a difference in prognosis between CK20-positive and CK20-negative PCa tumors (data not shown). However, the presence of CK20 positivity in PCa cases somewhat attenuated the poor prognostic effect of GP88 positivity when compared to GP88-positive tumors without CK20 positivity.

Furthermore, we examined the prognostic effects of GP88 positivity after stratification of the PCa patients according to their tumor stage and Gleason score. GP88 positivity was an independent prognostic factor for OS in the pT2 group but not in the pT3/pT4 group. This could be helpful in distinguishing patients with a shorter OS in this otherwise rather good prognostic pT2 group. The Gleason score groups were GS6, GS7a, GS7b, GS8 and GS9-10. In the GS7b group, all patients with GP88-negative tumors had improved survival (OS and DSS) in comparison to that of patients with GP88-positive tumors (Kaplan–Meier analysis). In the GS8 group, patients with GP88 positivity showed a poorer RFS compared to the RFS of those with GP88-negative tumors. However, in the multivariate analysis, GP88 positivity did not remain an independent prognostic factor, which could be explained by the rather small sizes of these subgroups.

How could GP88 affect OS, DSS and RFS?

GP88 has been shown to be involved in proliferation, survival, migration, angiogenesis, invasion, and matrix metalloprotease (MMP) activity [20]. It can increase the production of several matrix-degrading enzymes, such as MMP 13 and MMP 17, in adrenal carcinoma cells and in this way, regulates invasion and cell survival [21]. The prostate cancer cell lines PC-3 and DU145 show high RNA expression for MMP17 [22]. In addition, elevated MMP13 (and MMP11) protein expression in association with shorter RFS has been reported in PCa [23]. In addition, GP88 stimulates c-myc phosphorylation by affecting mitogen-activated protein kinase (MAP kinase ERK1/2) and the phosphatidyl-inositol-3-kinase (PI3K) pathways and stimulates src phosphorylation in breast cancer cells [24]. In PCa, signaling via the PI3K/AKT pathway suppresses androgen receptor-mediated gene expression and promotes androgen-independent cell growth [25,26,27]. Recently, elevated levels of phosphorylated ERK1/2 were detected in castration-resistant prostate cancer compared to levels in untreated primary prostate cancer. In addition, the presence of detectable phosphorylated ERK1/2 in the primary tumor was associated with biochemical failure, i.e., RFS, after radical prostatectomy independent of clinico-pathologic features [28]. An upregulation of nuclear c-myc protein expression has been described as an early oncogenic alteration in PCa, and overexpression of the c-myc gene has been associated with a shorter RFS [29]. An increase in c-src activity was detected in the tissue of androgen-independent prostate cancer (AIPC) compared to activity in androgen-sensitive prostate cancers (ASPC), and AIPC patients had a significantly shorter OS than AIPC patients without increased C-SRC activity [30]. Altogether, through its multiple effects on different PCa-relevant oncogenes/pathways, i.e., ERK1/2, PI3K, c-myc and c-src, GP88 appears to be an interesting target in PCa treatment. Antisense GP88 treatment of MDA-MB-468 breast cancer cells in a mouse model has been performed, resulting in a reduced number and weight of tumors compared to those of mice injected with untreated control cells [31]. Buraschi et al. used shRNA models to deplete endogenous progranulin from urothelial cancer cells. Progranulin depletion severely inhibited the migration, invasion and anchorage-independent growth of urothelial cancer cells. Interestingly, GP88 levels also correlated with response to cisplatin treatment in urothelial cancer [32]. In human HCC orthotopic xenograft models, GP88 antibody treatment was capable of inhibiting tumor growth. In addition, the combination of a GP88 antibody with a high dose of cisplatin resulted in the eradication of established intrahepatic tumors [33]. The application of GP88 antisense oligonucleotides or GP88 antibodies has not yet been tested in PCa models, but it might be a future option for PCa therapy. In summary, GP88 positivity appeared to be an independent prognostic factor in OS, DSS and RFS for all PCa patients and for younger PCa patients (≤65 years). In addition, in PCa patients with CK20-negative staining (IRS < 2), GP88 positivity was an independent parameter for OS and DSS. Considering that we have found, in a separate study [19], that serum GP88 levels were also associated with clinical outcomes, particularly in younger PCa patients, the combination of GP88 measurements in tumor biopsies and in serum would provide cost-effective assays to improve the management of PCa patients. Future studies measuring serum and tissue GP88 in PCa patients would allow us to explore this possibility.

## 4. Material and Methods

### 4.1. Patients and Tumor Material

The tissue microarrays (TMAs) comprised consecutively collected, formalin-fixed and paraffin embedded tumor samples of 442 PCa patients diagnosed in the Department of Pathology, University Hospital Erlangen from 1999 to 2010. The tumors originated from prostatectomies and the follow-up time from prostatectomy was 0 to 246 months with a median of 95 months. The tumor histology was reviewed by experienced uropathologists (AH and ME). All procedures were performed in accordance with the ethical standards established in the 1964 Declaration of Helsinki and its later amendments. All patients beginning in 2008 gave informed consent. For samples collected prior to 2008, the Ethics Committee in Erlangen waived the need for informed individual consent. The study is based on the approval of the Ethics Committee of the University Hospital Erlangen (No. 3755). As we used archival tissue several years after resection, analysis of GP88 protein expression was retrospective and did not affect clinical decisions. An overview of the clinico-pathologic parameters of the patients is given in Table 1. The PSA level precedes the prostatectomy.

### 4.2. Immunohistochemistry

CK20 expression was examined with a monoclonal mouse CK20 antibody (Dako, Hamburg, Germany, Clone Ks 20.8, dilution 1:50) by a routine staining procedure as previously described [34]. For the GP88 protein expression study, a manual immunohistochemistry (IHC) protocol was applied as previously described [35]. Briefly, after heat pretreatment at 120 °C for 5 min with TE buffer, pH 9, and peroxidase blocking (Dako), a primary antibody against GP88 (monoclonal mouse anti-GP88/PGRN antibody, Cat. No. AG10008; Precision Antibody, A&G Pharmaceutical, Columbia, MD, USA) was applied for 30 min. The stained specimens were viewed at an objective magnification of ×100 and ×200. The expression of GP88 was detected in the cytoplasm by assessing the percentage of stained tumor cells and the staining intensity semi-quantitatively. The percentage of positive cells was scored as follows: 1, 1%–9% positive cells; 2, 10%–50%; 3, 51%–80%; and 4, >80% positive cells. Staining intensity was scored as 0, negative; 1, weak; 2, moderate; and 3, strong. The immunoreactive score (IRS) was calculated as the product of staining percentage and staining intensity, resulting in an IRS from 0 to 12. Their product resulted in an immunoreactive score (IRS) from 0 to 12 [36]. GP88 was not detected in the normal tissue adjacent to the tumor. Negative control slides without the addition of primary antibody were included for each staining experiment. For the IHC analysis, the patients were grouped by IRS < 2 and IRS ≥ 2 for GP88 and CK20 staining. Slides were scanned with a P250 slide scanner (3DHistech, Budapest, Hungary) and analyzed using CaseViewer2.0 (3DHistech).

### 4.3. Statistical Analyses

The correlations between the IHC scores and clinico-pathological data were calculated using Spearman’s bivariate correlation or the Chi2-test. The associations of the expression of GP88 with overall survival (OS), disease-specific survival (DSS), and relapse-free survival (RFS) were determined by univariate (Kaplan–Meier analysis and Cox’s regression hazard models) and multivariate analyses (Cox’s regression hazard models, adjusted for age, tumor stage, and Gleason score). We tested the proportionality assumption by plotting the log(-log(survival)) versus log of survival time. The resulting graphs show parallel lines. Following standard practice in retrospective survival analysis, the common time point zero of all patients was the date of the tumor surgery. A *p*-value < 0.05 was considered statistically significant. The statistical analyses were performed with the SPSS 21.0 software package (SPSS Inc., Chicago, IL, USA).

## 5. Conclusions

The growth and survival factor progranulin (GP88) is an independent prognostic factor for overall, disease-specific and relapse-free survival for prostate cancer (PCa) patients.

Especially in a subgroup of younger PCa patients (≤65 years), elevated GP88 protein expression was significantly associated with shorter overall, tumor-specific and relapse-free survival. This finding has an impact against the background of increasing numbers of PCa cases among younger men below 65 years of age.

## Figures and Tables

**Figure 1 cancers-11-02029-f001:**
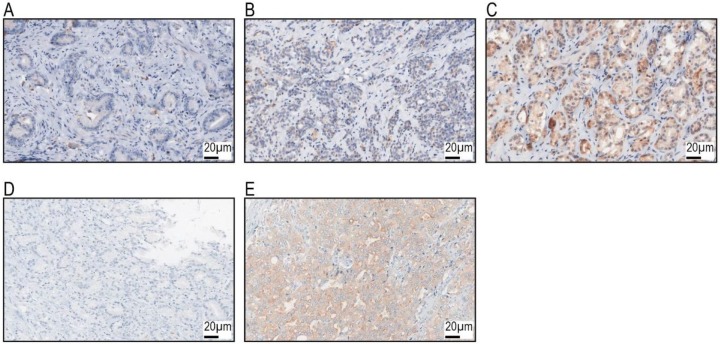
GP88 and CK20 immunohistochemical staining upper row GP88 staining; (**A**): IRS = 0; (**B**): IRS = 2 (Intensity weak:1 and percentage 30%: 2); (**C**): IRS = 8 (intensity moderate: 2 and percentage: 100%: 4) lower row CK20 staining; (**D**): IRS = 0; (**E**): IRS = 8 (intensity moderate: 2 and percentage: 100%: 4). All photos are in a 40× magnification.

**Figure 2 cancers-11-02029-f002:**
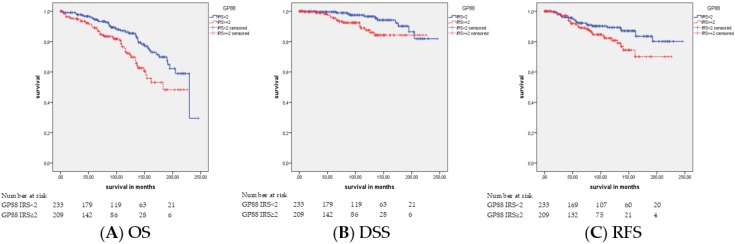
Kaplan–Meier analyses: Association of GP88 staining with the prognosis in all PCa patients. GP88 protein expression was associated with (**A**) OS (*p* = 0.002), (**B**) DSS (*p* = 0.018) and (**C**) RFS (*p* = 0.040; all log rank test).

**Figure 3 cancers-11-02029-f003:**
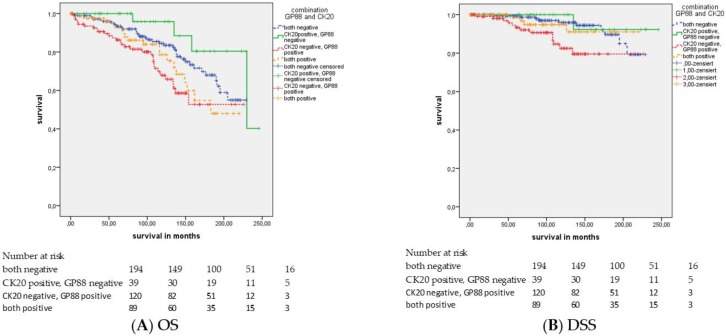
Kaplan–Meier analysis: Association of the combination of GP88 staining and CK20 staining with the prognosis of all PCa patients. The combination of GP88 (IRS < 2 vs. IRS ≥ 2) and CK20 (IRS < 2 vs. IRS ≥ 2) protein staining resulted in four groups: group 0: both markers are negative; group 1: CK20 positive and GP88 negative; group 2: CK20 negative and GP88 positive; and group 3: both markers are positive. For (**A**) OS (*p* = 0.005) and (**B**) DSS (*p* = 0.015): The best survival was exhibited by patients in group 1, the second best survival in group 0, the third best survival in group 3, and the worst survival appeared in group 2.

**Table 1 cancers-11-02029-t001:** Clinico-pathological and immunohistochemical data of the PCa patients.

	N
All PCa Patients	442
Age median in years (range) (IQR)	65 (45–83) (61–69)
Pathological tumor stage (pT)	
pT2	260
pT3	151
pT4	31
Gleason score (GS)	
GS 6	237
GS 7a	82
GS 7b	32
GS 8	27
GS 9–10	34
GS unknown	30
PSA at prostatectomy (median) (IQR)	(3.54) (0.83–7.46)
<4 ng/mL	194
≥4 ng/mL	180
unknown	68
GP88 staining (median) (IQR)	(1.5) (0–4)
IRS < 2	233
IRS ≥ 2	209
CK20 staining (median) (IQR)	(0) (0–2)
IRS < 2	314
IRS ≥ 2	128

Abbreviation: IQR: interquartile range.

**Table 2 cancers-11-02029-t002:** Survival and immunohistochemical data of the PCa patients.

Clinical endpoint		GP88 IRS < 2	GP88 IRS ≥ 2		CK20 IRS < 2	CK20 IRS ≥ 2
Overall survival (OS)	N			N		
alive	350	191	159	350	245	105
dead	92	42	50	92	69	23
Disease-specific survival (DSS)						
Yes	414	222	192	414	291	123
No	28	11	17	28	23	5
Relapse-free survival (RFS)						
Yes	389	209	180	389	279	110
No	53	24	29	53	35	18

**Table 3 cancers-11-02029-t003:** Kaplan-Meier analysis: Association of GP88 staining with mean OS, mean DSS or mean RFS.

			Kaplan–Meier Analysis				
GP88	N	OS		DSS		RFS	
IRS ≥ 2 vs. IRS < 2							
		Months	*p*	Months	*p*	Months	*p*
All patients	442	164.9 vs. 195.8	0.002	204.3 vs. 231.1	0.018	186.4 vs. 218.1	0.040
Tumor stage pT2	260	159.9 vs. 193.4	0.020	196.1 vs. 220.7	0.021		n.s.
GS7b (GS8*)	32	n.d.	0.004	n.d.	0.029	99.0 vs. 201.8	0.008
Age ≤ 65 years	230	169.5 vs. 227.3	0.001	186.1 vs. 237.1	0.003	153.1 vs. 222.8	0.004
CK20 IRS < 2	314	162.9 vs. 186.8	0.009	197.5 vs. 216.1	0.005		n.s.

GS8*—Gleason score 8 group for RFS (Gleason score 7b group for OS and DSS). n.d.—not determined, since all patients with GP88 < IRS2 survived in OS and DSS (both N = 13).

**Table 4 cancers-11-02029-t004:** Univariate and Multivariate Cox’s regression analyses: Association of GP88 staining with OS, DSS and RFS.

			Univariate Cox’s Regression Analysis					
GP88	N	OS		DSS		N	RFS	
IRS ≥ 2 vs. IRS < 2								
		HR (95% CI)	*p*	HR (95% CI)	*p*		HR (95% CI)	*p*
All patients	442	1.9 (1.2–2.9)	0.003	2.5 (1.1–5.3)	0.021	442	1.7 (1.0–3.0)	0.043
Tumor stage pT2	260	2.1 (1.1–3.9)	0.022	4.6 (1.2–18.4)	0.030	260		n.s.
GS7b (GS8*)	32		n.s.		n.s.	27	10.2 (1.2–86.4)	0.033
Age ≤ 65 years	230	3.6 (1.6–8.3)	0.002	5.5 (1.6–18.9)	0.006	230	2.9 (1.4–6.4)	0.005
CK20 IRS < 2	314	1.9 (1.2–3.1)	0.010	3.2 (1.4–7.5)	0.007	314		n.s.
			**Multivariate Cox’s Regression Analysis**					
GP88	**N**	**OS**		**DSS**		**N**	**RFS**	
IRS ≥ 2 vs. IRS < 2								
		**HR (95% CI)**	***p***	**HR (95% CI)**	***p***		**HR (95% CI)**	***p***
All patients	412	1.8 (1.1–2.7)	0.011	2.4 (1.0–5.5)	0.039	412	1.7 (1.0–3.1)	0.046
Tumor stage pT2	242	1.9 (1.0–3.8)	0.043		n.s.	242		n.s.
GS7b (GS8*)	32		n.s.		n.s.	27		n.s.
Age ≤ 65 years	215	3.8 (1.5–9.6)	0.004	6.0 (1.6–22.7)	0.008	215	3.7 (1.6–8.8)	0.003
CK20 IRS < 2	296	1.8 (1.1–3.0)	0.018	2.8 (1.1–7.0)	0.028	296		n.s.

GS8*—Gleason score 8 group for RFS (Gleason score 7b group for OS and DSS).

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
