# Peer review of "Expression of GP88 (Progranulin) Protein Is an Independent Prognostic Factor in Prostate Cancer Patients"

_cancers, 2019, doi:10.3390/cancers11122029_

Round 1

Reviewer 1 Report

The paper was overall well written and it will be certainly of great interest for the readers. The Authors are to be congratulated for their effort.

In order to make the manuscript more appealing for publication I would ask the author to make the following amendments, in particular regarding the statistical analyses carried out and the presentation of the results.

In Table 1, OS, DSS and RFS are not clinical and pathological variables, but prognostic variables. I suggest to separate these from this table and better explaining in another table. In table 1, please specify 2 patients missing for pN. I suggest to insert in Table 1 also the PSA value as a descriptive analysis. Moreover it should be more described the population considered for the analysis. The “PSA value at prostatectomy” referred to the value that precedes the prostatectomy or the first PSA after prostatectomy? Please specify in the material and methods section. Since the authors found a significant positive correlation between GP88 IRS and the PSA, why did not they insert the PSA as covariate in the multivariate cox model? For the univariate and the multivariate Cox models, did the authors verify if the proportional hazard assumption was respected? Please, specify in the text. Since Cox proportional hazard models were performed, please correct all the "RR" in "HR". At the end of line 107, please add the term “respectively” in order to better describe the associations. Kaplan-Meier analysis could be more interpretable if in the table had been reported the medians and the inter-quartile ranges for the considered variables. Please specify in the table 2 header and in supplementary Table 4 header the measure of the time (mean, or medians). Line 109: GP88 was an independent “prognostic” factor because it is associated to the overall survival, it was not a “predictive” factor. It is not clear how the Cox regression models were built: a multivariate model should be create with all the patients that have the covariate variable not missing. So that, the model have a define and unique N, while in the Table 3 and in Suppl. Table 5 (multivariate) the N changes. Moreover it is not clear if these analyses were performed on the all PCa patients or on a subgroup of patients (only patients with pT2 stage, or only patients with GS7b-GS8, or only in age<=65 years). At line 124 “adjusted for the Gleason score and age”, it is not adjusted also for CK20 IRS? The same at lines 156-157. To declare that the GP88 is an independent prognostic factor, the multivariate analysis have to include also the GP88 and it have to be built for the all PCa patients, first. In a second time it will be confirmed on a sub-group analysis. Line 139: it was not reported the HR of the Gleason Score in the multivariate table 3, so it is not possible to assess that it is not an independent prognostic factor. All the figures (excluded figure 1) are not clearly readable because the legends were wrote in a too little font, it is impossible to recognize the different lines in a black and white version, please improve the graphic design. It could be useful to insert under the graph the number of patients at risk. Lines 122-123, please specify in the text that the association of GP88 and risk of death referred to tumor stage pT2. Line 134, please add the mean or median time. Lines 162-165: please move this concept in the introduction section. Lines 165-166: this was already described before, and it is not necessary to repeat it. Line 311: in the statistical analysis section miss the CK20 as covariate variable for adjusting Cox proportional hazard model. In Suppl. Table 3A: pay attention to the graphical because “GP88 IRS” are not in the same line and this makes misleading  the reading.

Reviewer 2 Report

Table 1 (page 82) Before line "pT" write "Tumor stage group". Trancation of the abberviation in line 117 only. It is not clear what is abbreviated to "pN". This needs to be explained. The text of the results does not analyze GP88 association with pN. Do you need to include pN in the table? The text of "Results"analyze GP88 correlation with PSA value at prostatectomy.  But the section "Materials and Methods" doesn't mention how PSA was obtained - from the medical records, the registry? The term "tumor stage groups" does not use the same abbreviation throughout the text: 117 line (pT2.....) 208, 209 line - (T1/T2) 236 line - (pT3/4) Please use one obbreviation!

Author Response

Please, see attachement.

Reviewer 3 Report

Overview and general recommendation: The present manuscript is to characterize if GP88 protein can be used as an independent prognostic biomarker in prostate cancer patients. GP88 is a widely expressed glycoprotein with pleiotropic function. It has been linked to a host of physiological processes and diverse pathological states. Currently, various GP88-targeted approaches are emerging as attractive therapeutic interventions in a broad spectrum of diseases including cancer, inflammatory diseases, neurological disorders, injury, tissue regeneration, and some rare diseases such as lysosomal diseases. Few studies focus on its prognostic utility. So this topic is timely. The authors uses IHC staining on prostate cancer tissue microarrays and analyzed the correlation between GP88 and prostate cancer progression. They found a significant positive correlation of GP88 with PAS value and cytokeratin 20 immunoreactive score, further, GP88 protein expression correlated with overall disease-specific and relapse-free survival and this correlation only happens in younger (<65) patients and negative CK20 subgroup patients. This is an interesting result. But why this phenomenon was not in older (>65) patients and/or CK20 positive patients need to be justified. This manuscript is overall well written and I believe it will be valuable to Oncologists, Urologists, and Prostate Cancer Researcher.

Major Comments:

The authors separate the prostate cancer samples into CK20 positive and negative groups, this should be justified. What’s the relationship between GP88 and CK20 proteins? The authors should give some introduction about CK20 in the introduction section and discuss the utility of CK20 and why GP88 expression is different in CK20 negative and positive samples.

Minor Comments:

The short name of ‘prostatic intraepithelial neoplasia’ is PIN, not prostate intraepithelial lesion. 1, the authors only provide low power pictures for the IHC results, each IRS results should provide a high power picture to indicate the cytoplasmic staining. Figure legend is too simple and can’t self-standard. The author should provide a detailed calculation of the IRS score and analysis. How do you exclude the nuclear staining when analyzing the IRS score? Check the methods: how to exclude nuclear staining? IRS =8 looks like it has nuclear staining. provide high power pictures to show the exact positive staining results.

Author Response

Please, see attachment.

Round 2

Reviewer 1 Report

The majority of the corrections was implemented except for the figures requested.

The legend of the figures is as important as the figures themselves and the authors have not increased the font of the legend despite it having been requested. The figures are very interesting, but should be made readable.